# Photo-tailored heterocrystalline covalent organic framework membranes for organics separation

Jinqiu Yuan [1,2,7], Xinda You[1,2,7], Niaz Ali Khan [1,2], Runlai Li [3], Runnan Zhang [1,2,4], Jianliang Shen [1,2], Li Cao[1,2], Mengying Long [1,2,5], Yanan Liu[1,2], Zijian Xu[1,2], Hong Wu [1,2,6✉] & Zhongyi Jiang [1,2,4,5✉]

Organics separation for purifying and recycling environment-detrimental solvents is essential to sustainable chemical industries. Covalent organic framework (COF) membranes hold great promise in affording precise and fast organics separation. Nonetheless, how to well coordinate facile processing—high crystalline structure—high separation performance remains a critical issue and a grand challenge. Herein, we propose a concept of heterocrystalline membrane which comprises high-crystalline regions and low-crystalline regions. The heterocrystalline COF membranes are fabricated by a two-step procedure, i.e., dark reaction for the construction of high-crystalline regions followed by photo reaction for the construction of low-crystalline regions, thus linking the high-crystalline regions tightly and flexibly, blocking the defect in high-crystalline regions. Accordingly, the COF membrane exhibits sharp molecular sieving properties with high organic solvent permeance up to 44-times higher than the state-of-the-art membranes.

[1] Key Laboratory for Green Chemical Technology of Ministry of Education, School of Chemical Engineering and Technology, Tianjin University, Tianjin 300072, China. [2] Collaborative Innovation Center of Chemical Science and Engineering (Tianjin), Tianjin 300072, China. [3] Department of Chemistry, National University of Singapore, 3 Science Drive 3, Singapore 117543, Singapore. [4] Zhejiang Institute of Tianjin University, Ningbo, Zhejiang 315201, China. [5] Joint School of National University of Singapore and Tianjin University, International Campus of Tianjin University, Binhai New City, Fuzhou 350207, China. [6] Tianjin Key Laboratory of Membrane Science and Desalination Technology, Tianjin University, Tianjin 300072, China. [7] These authors contributed equally: Jinqiu Yuan, Xinda You. ✉email: wuhong@tju.edu.cn; zhyjiang@tju.edu.cn

Global demand for organic solvents is expected to surpass 20 million tons by 2023 for use in chemical, fragrance, and pharmaceutical processes due to the solubility and product selectivity requirements. These solvents must be separated from intermediate chemicals or products for recycling and minimizing emission[1–4]. Pressure-driven membrane separation operating at mild conditions without phase change holds great promise due to its much lower energy consumption and organic vapor emission compared with the commonly used thermal-based distillation technology[5–8]. However, conventional membranes which are fabricated from network polymers are often difficult to simultaneously achieve high solvent resistance and fast molecular sieving performance[9,10], necessitating new-generation membrane materials with highly robust, ordered, and porous structures.

Two-dimensional covalent organic framework (COF) is an emerging crystalline framework material comprising rigid molecular building blocks connected by robust in-plane covalent bonds and out-of-plane π-π interactions, exhibiting excellent solvent-resistant properties[11–14]. Moreover, the COF material features an ordered porous structure that has up to 100 times higher surface area than most network polymers[15]. Theoretically, high-crystalline COF membranes (COMs) formed totally by rigid crystallites can harvest high solute selectivity and possible high solvent permeability, however, the defect-free membranes are quite difficult to fabricate[16–18]. Low-crystalline COMs are much easier to fabricate, but often accompany with much less ordered structure and a dramatic sacrifice in solute selectivity as well as possible solvent permeability[19,20]. It proves a grand challenge to acquire good membrane processability and sufficiently high crystallinity through adjusting the crystallinity of the whole COM. In this case, the COM can be regarded as the homocrystalline COM.

Herein, we propose a concept of heterocrystalline membrane comprised of both high-crystalline regions and low-crystalline regions. The heterocrystalline COM is prepared through a two-step procedure based on sequential Schiff-base reactions where the bond linkage can tautomerize under photo irradiation. The first step is dark reaction to generate the high-crystalline COM. The second step is photo reaction to generate low-crystalline regions in the intercrystalline defect of the COM, thus linking the high-crystalline regions tightly and flexibly, blocking the defect in high-crystalline regions. Photochemistry, which can interfere in proton transfer, is a widely-used and green way to modulate chemical reactions[21]. It has been demonstrated that photo-induced excited-state intramolecular proton transfer (ESIPT) can enable enol-imine to keto-enamine tautomerization[22,23], while the enol-imine linkage is the key for the commonly used Schiff-base COF to form high-crystalline structure[24]. This photo-induced tautomerization provides the chemical basis to control the reactive-crystallization procedure of COF by photo irradiation. In this study, we present a photo-tailoring strategy to prepare the heterocrystalline COM (Fig. 1). Under dark settings, the reversible enol-imine linkage can break up and reform, thus correcting the initial mismatched amorphous structure, allowing the formation of a high-crystalline COM. Subsequently, photo irradiation is introduced and the phototautomerization of enol-imine linkage inhibits the "error-correcting" process, allowing the formation of low-crystalline regions in the intercrystalline defects of the COM. By tuning the photo reaction time, the low-crystalline regions can tightly and flexibly link the high-crystalline regions to obtain the defect-free COM. Meanwhile, the well-preserved high-crystalline regions with ordered porous structures would enable precise and fast organics separation. The resulting COM displayed sharp molecular sieving properties, as manifested by the high organic solvents permeance up to 44-times higher than the state-of-the-art membranes.

## Results

**Photo-tailored reactive-crystallization of Schiff-base COM.** The synthetic routes of COMs by either dark reaction or photo reaction were illustrated in Fig. 2a. Initially, precursor trialdehyde (Tp) and diamine (Bpy) would polymerize into an amorphous network via enol-imine linkage[25]. During dark reaction, the reversible enol-imine linkage breaks and reforms slowly, thus converting the initial amorphous network into the thermodynamically stable crystalline framework as a result of the "error-correcting" process[26]. Then, the enol-imine linkage tautomerizes irreversibly to stable keto-enamine form because the basicity of three imine nitrogens ($C = N$) dominates over the aromaticity of the central benzene ring[27,28]. The optimal reaction time was set at 96 h, and the Bpy and Tp concentrations were set at 0.30 and 0.20 mmol $L^{-1}$, respectively (Supplementary Fig. 2, 3). Fourier transform infrared (FTIR), solid-state $^{13}C$ nuclear magnetic resonance (NMR) and X-ray photoelectron spectrometer (XPS) spectra confirm the formation of the keto-enamine-linked COM by dark reaction (DCOM), as indicated by the $C = C$ stretching band at ca. 1566 $cm^{-1}$ (Supplementary Fig. 4), enamine carbon resonance at 151.0 ppm (Supplementary Fig. 5), and secondary amine (=C-NH) with binding energy of 399.8 eV (Supplementary Fig. 6). X-ray diffraction (XRD) pattern suggests the high crystallinity of the DCOM, which shows an intense and sharp peak at ~3.5° corresponding to the reflection from the 100 crystal plane (Fig. 2b)[29].

The light source of photo reaction is supplied by a xenon lamp ($\lambda = 200-400$ nm) with an irradiation intensity from 1.5 to 9.0 mW $cm^{-2}$. During photo reaction, the initially formed reversible enol-imine linkage tautomerizes rapidly into keto-enamine form via photo-induced ESIPT (Fig. 2a, c, $k_{ESIPT} > 10^{12} s^{-1}$)[22,23], resulting in a decrease of reversible enol-imine linkage and thus inhibiting the "error-correcting" process. This photo-induced tautomerization from enol-imine to keto-enamine is confirmed by the steady-state photoluminescence emission spectrum. Figure 2d shows the fluorescence properties of the initial amorphous material dispersed in either the aqueous phase or the organic phase of interfacial polymerization. A dual fluorescence emission phenomenon is observed, with the wavelength emission at 400–450 nm representing the enol-imine form (normal emission) and the wavelength emission at 580–630 nm reflecting the keto-enamine tautomeric form (ESIPT emission)[30,31], confirming photo-induced enol-imine to keto-enamine tautomerization. The COMs formed by photo reaction (PCOMs) display a substantially weaker and wider (100) diffraction peak than DCOM (Fig. 2b), even under low irradiation intensity (1.5 mW $cm^{-2}$), indicating the pronounced influence of photo irradiation on the COM crystallization. The crystallinity of the PCOMs decreases with the increase of the irradiation intensity (Fig. 2b), offering a facile approach to control the crystalline structure. NMR spectra reveal that the resonance peaks of PCOM are wider and less resolved than those of DCOM, confirming the poor development of crystalline structure in PCOM (Supplementary Fig. 5)[32]. FTIR spectra demonstrate that the $C = C$ stretching band in the keto-enamine linkage of PCOMs was more intense than that of DCOM (Supplementary Fig. 4), ascribing to the small energy barriers of enol-keto phototautomerization[33,34]. These findings demonstrate a simple and effective strategy for tailoring the crystalline structure of Tp-Byp COM. To evaluate the generality of this strategy, we further prepared two kinds of Schiff-base COM, Tp-Tta and Tp-Azo. It has been found that both the Tp-Tta and Tp-Azo COM formed by photo reaction exhibit notably less crystallinity than those formed by dark reaction (Supplementary Fig. 7). This strategy offers the possibility to tailor heterocrystalline COM by controlling the dark and photo reactions during membrane formation.

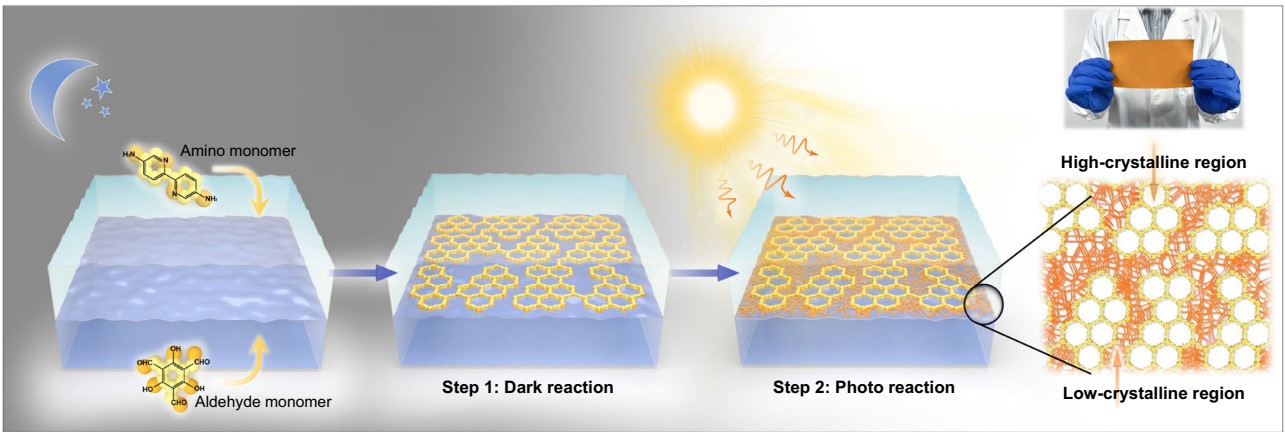

**Fig. 1 Schematic preparation of heterocrystalline COM.** Preparation of heterocrystalline COM by subsequent dark reaction and photo reaction using interfacial polymerization, where the top light-blue layer is the aqueous phase and the bottom navy-blue layer is the organic phase. 1,3,5-triformylphloroglucinol (Tp) and 2,2′-bipyridine-5,5′-diamine (Bpy) are used as aldehyde monomer and amino monomer, respectively, to prepare Schiff-base Tp-Bpy COM. The inset digital photograph is the photo-tailored heterocrystalline COM deposited on non-woven fabrics with the size of ~15 cm × 8 cm (Supplementary Fig. 1).

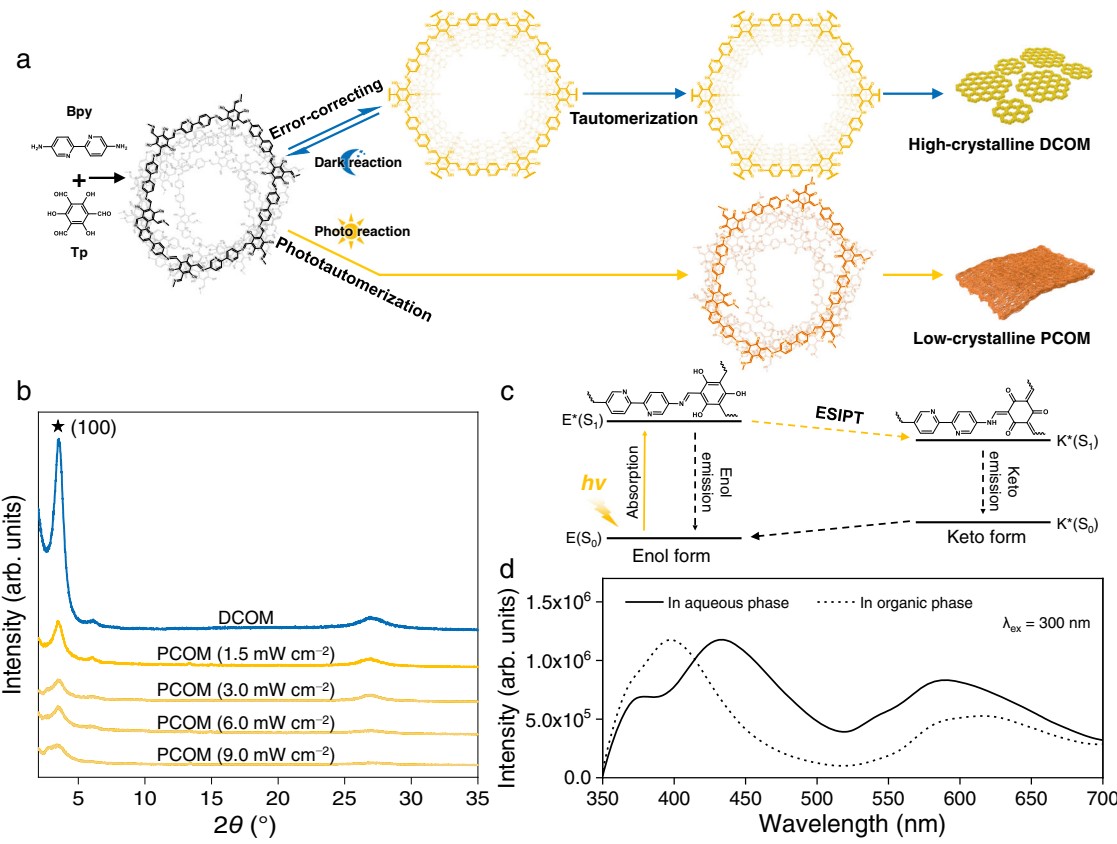

**Fig. 2 Photo-tailored reactive-crystallization of Schiff-base COM. a** Synthetic route (blue) of COM by dark reaction (DCOM) and synthetic route (yellow) of COM by photo reaction (PCOM) using reactive monomers of 2,2′-bipyridine-5,5′-diamine (Bpy) and 1,3,5-triformylphloroglucinol (Tp). **b** XRD patterns of the DCOM (blue) and PCOMs (yellow) formed under varied irradiation intensity ranging from 1.5 to 9.0 mW cm$^{-2}$. **c** Schematic illustration of the excited-state intramolecular proton transfer (ESIPT) process of enol-imine linkage. **d** Steady-state photoluminescence emission spectra of the initial amorphous material dispersed in either aqueous phase (solid lines) or organic phase (dashed lines). These samples were excited at 300 nm.

**Preparation and characterizations of DPCOMs.** The morphologies of DCOM formed under dark condition and PCOMs formed under varied irradiation intensity are systematically investigated. For DCOM, fiber-crystallite assembled morphology with tens-of-nanometer-sized intercrystalline defects can be observed (Fig. 3a, Supplementary Fig. 8), revealing poor processibility of high-crystalline COM. In contrast, the morphology of PCOMs changes into a flexible polymer-like structure with inappreciable intercrystalline defects by increasing the irradiation intensity from 1.5 to 9.0 mW cm$^{-2}$ (Fig. 3a, Supplementary

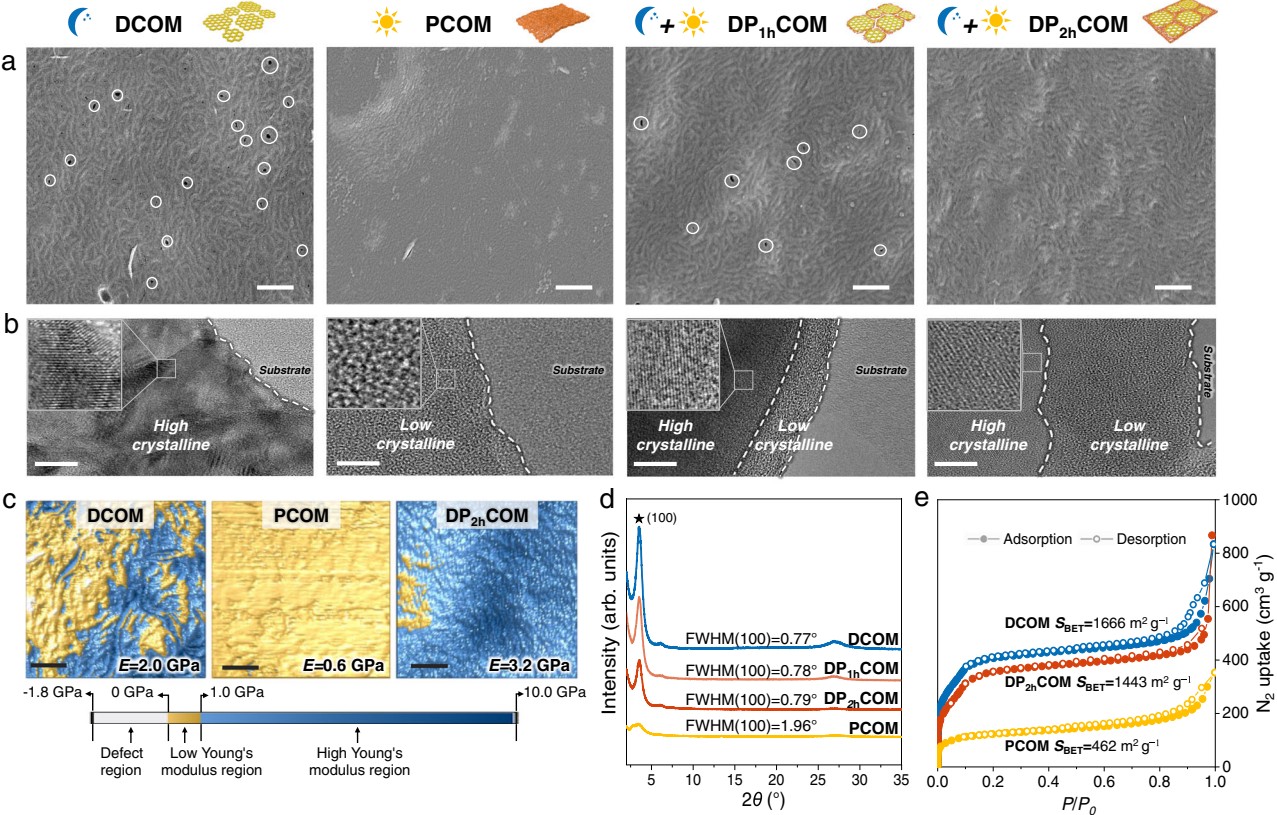

**Fig. 3 Structure and morphology characterizations. a** Top-view SEM images of COMs deposited on track-etched substrate membranes. The intercrystalline defects are marked by the white circle. Scale bar = 500 nm. **b** TEM images of the COMs. Insets are high-magnification images where the high-crystalline regions show lattice diffraction patterns. Scale bar = 10 nm. **c** Young's moduli of COMs tested using the peak force quantitative nano-mechanical property mapping method. Scale bar = 200 nm. **d, e** XRD patterns (**d**) and $N_2$ adsorption isotherms (**e**) of DCOM (blue), DPCOMs (red), and PCOM (yellow). The PCOM here is fabricated under 9.0-mW $cm^{-2}$ irradiation.

Fig. 9), manifesting the superior processability of low-crystalline COM formed under 9.0-mW $cm^{-2}$ irradiation intensity.

The heterocrystalline COM, denoted as DPCOM, was fabricated by dark reaction first in the same way as DCOM, followed by photo reaction under 9.0-mW $cm^{-2}$ irradiation (Fig. 1). After photo reaction, the successful incorporation of low-crystalline regions into DPCOM is revealed by FTIR analysis, where the C=C (1566 $cm^{-1}$) of keto-enamine linkage gradually increases with the increase of photo reaction time (Supplementary Fig. 10). Transmission electron microscope (TEM) analysis indicates that the low-crystalline regions grow in the edge of the high-crystalline regions bearing lattice diffraction pattern and the growth area is proportional to photo reaction time (Fig. 3b). After forming low-crystalline regions, the size and quantity of the intercrystalline defects within $DP_{2h}COM$ observably decrease (Fig. 3a, Supplementary Fig. 8). Moreover, the thickness of the $DP_{2h}COM$ does not increase (~55 nm), suggesting that the low-crystalline regions grow in the intercrystalline defects instead of along the thickness of the membrane (Supplementary Fig. 11). This is ascribed to the inherent self-inhibition effect in the interfacial polymerization process, where the monomers dissolved separately in organic and aqueous phase prefer to infuse and polymerize in the defects of interfacial membrane[35]. We also use atomic force microscope (AFM) tip to determine the local Young's modulus of the individual regions of COMs (Fig. 3c). The $DP_{2h}COM$ (3.2 GPa) exhibits much higher average modulus than that of PCOM (0.6 GPa) and DCOM (2.0 GPa) due to the rigid high-crystalline regions with large modulus and the decreased defect regions bearing tiny modulus. These results

demonstrate that the intercrystalline defects of $DP_{2h}COM$ are effectively sealed by the low-crystalline regions. Furthermore, our strategy can even seal the several-hundred-nanometer defects of COM (Supplementary Fig. 12).

The photo irradiation would not affect the crystalline structure of high-crystalline regions, which can be proved by the almost unchanged full width at half maximum (FWHM) of the (100) diffraction peak of DPCOM (Fig. 3d)[36]. This high crystallinity endows $DP_{2h}COM$ with a very porous structure, which shows a Brunauer–Emmett–Teller surface area ($S_{BET}$) of up to 1443 $m^2 g^{-1}$ (Fig. 3e). This value is slightly lower than that of DCOM due to the incorporation of low-crystalline regions but an order of magnitude higher than previously reported organics separation membranes (Supplementary Table 1). The above results prove that the as-prepared heterocrystalline COMs can simultaneously satisfy the requirements for both membrane processibility and sufficient high crystallinity.

**Organics separation performance of DPCOMs.** The organics separation performance of COMs was evaluated in terms of dye rejection and organic solvent permeance. The DCOM bearing an intrinsic pore of 1.6 nm displays poor rejection (45%) to Evans blue with a dimension of 1.2 × 3.1 nm due to the tens-of-nanometer intercrystalline defects (Fig. 4a). The rejection of $DP_{2h}COM$ rises to 99%, as high as that of defect-free and dense PCOM prepared under 9.0-mW $cm^{-2}$ irradiation intensity (Fig. 4a, Supplementary Fig. 13), confirming that the generated low-crystalline regions seal the non-selective intercrystalline

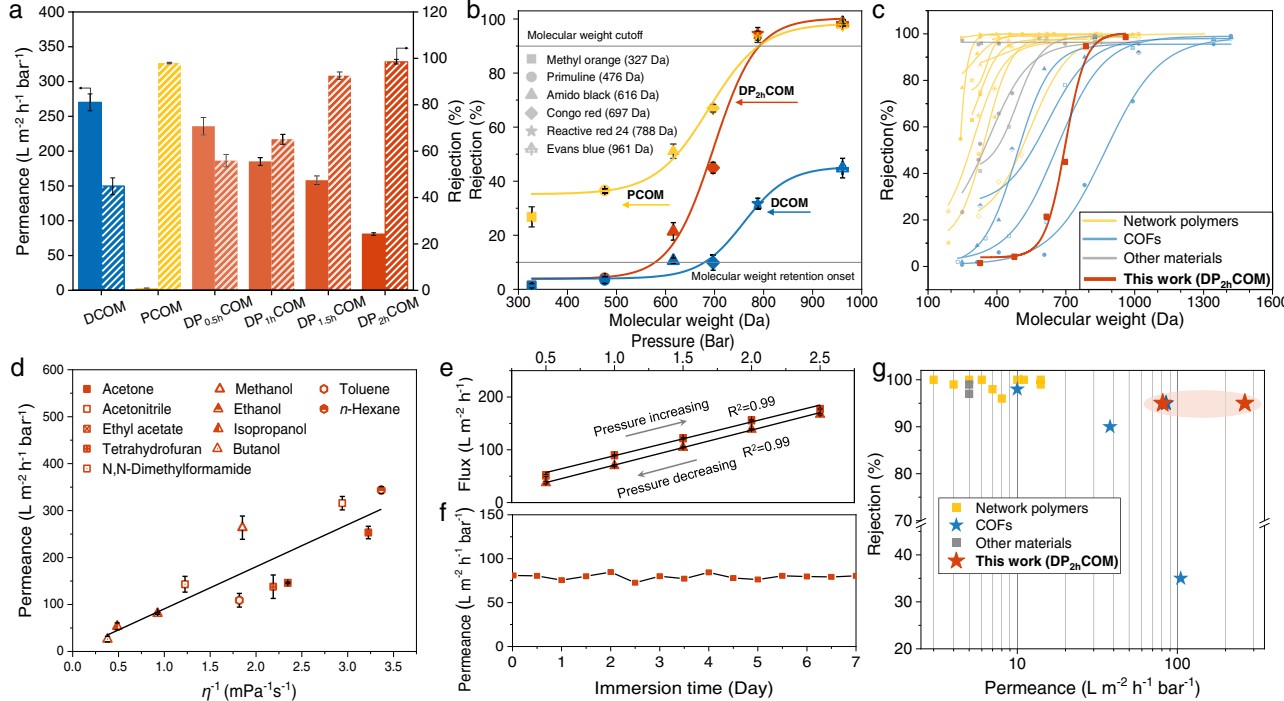

**Fig. 4 Organics separation performance. a** Pure ethanol permeance and Evans blue rejection of DCOM (blue), PCOM (yellow), and DPCOMs (red). **b** The rejection of dyes with different molecular weights through DCOM (blue), $DP_{2h}COM$ (red), and PCOM (yellow). The fitted sigmoidal model of the rejection curve is Doseresp. **c** Comparison of the rejection curve of $DP_{2h}COM$ with the reported state-of-the-art membranes. **d** Permeance of pure organic solvents through the $DP_{2h}COM$ as a function of their inverse viscosity. **e** Variation of the ethanol flux of $DP_{2h}COM$ under varying pressure. **f** Ethanol permeance of $DP_{2h}COM$ after solvent immersion. **g** Comparison of the dye (700–900 Da) rejection and solvent permeance (ethanol or methanol) of $DP_{2h}COM$ with the reported state-of-the-art membranes. The specific separation data of the reported membrane are listed in Supplementary Table 1, 3. The PCOM here is fabricated under 9.0-mW cm$^{-2}$ irradiation. All the error bars in this figure represent the average deviation ($n = 3$ independent experiments), data are presented as mean values ± SD. Dye concentration: 20 ppm for Evans blue, 50 ppm for other dyes.

defects effectively. We further investigated the rejections of $DP_{2h}COM$ to a series of dye solutes with various molecular weights (Supplementary Table 2). For all dyes, the adsorption on membranes is as low as 0.41 μg m$^{-2}$, which suggests that the rejection value is based on size exclusion rather than adsorption (Supplementary Fig. 14). As shown in Fig. 4b, the molecules with molecular weights >800 Da (reactive red 24 and Evans blue) can be effectively rejected by $DP_{2h}COM$ with rejections more than 90%, while molecules with molecular weights <600 Da (methyl orange and primuline) can easily permeate through the membrane with rejections less than 10%. The D-value of molecular weight cutoff (MWCO, 800 Da) and molecular weight retention onset (MWRO, 600 Da) is only 200 Da. The rejection curve of $DP_{2h}COM$ is much steeper than that of high-crystalline DCOM, low-crystalline PCOM, and other previously reported membranes attributing to its defect-free and crystalline ordered pore structure (Fig. 4c, Supplementary Table 3). Furthermore, a mixed dye separation experiment of methyl orange (327 Da) and Evans blue (961 Da) was conducted (Supplementary Fig. 15). The Evans blue can be completely rejected by $DP_{2h}COM$, while the methyl orange could pass through freely. This precise molecular sieving ability endows $DP_{2h}COM$ with prospective potential for separating organic mixtures.

The ethanol permeance of $DP_{2h}COM$ is as high as 81 L m$^{-2}$ h$^{-1}$ bar$^{-1}$, 26-times higher than that of PCOM with 3.6-times lower porosity, demonstrating the importance of porous structure in the organic molecules transport (Fig. 4a). The $DP_{2h}COM$ exhibits superior organophilic behavior (Supplementary Fig. 16), and then we further evaluated its permeation properties for different types of organic solvents including apolar (toluene, n-hexane), polar protic

(methanol, isopropanol, butanol), and polar aprotic (acetonitrile, acetone, ethyl acetate, dimethylformamide, tetrahydrofuran) solvents (Fig. 4d). The solvent permeance of $DP_{2h}COM$ is found to be linearly proportional to the inverse of solvent viscosity ($\eta^{-1}$). n-Hexane, with a viscosity of $2.97 \times 10^{-4}$ Pa·s, gives the highest permeance of 343 L m$^{-2}$ h$^{-1}$ bar$^{-1}$ (Supplementary Table 4). Methanol, the most used model solvent, with a small viscosity of $5.4 \times 10^{-4}$ Pa·s, also gives high permeances of 264 L m$^{-2}$ h$^{-1}$ bar$^{-1}$. The viscous flow behavior of the solvent through $DP_{2h}COM$ is ascribed to the solvent resistance and rigid pore structure[37,38]. Moreover, due to this structure, the solvent flux of $DP_{2h}COM$ increases linearly with an increase in the transmembrane pressure, revealing superior compaction resistance of $DP_{2h}COM$ (Fig. 4e). And the permeance keeps constant even after 7-day solvent immersion (Fig. 4f, Supplementary Fig. 17).

The solute rejection and solvent permeance are compared with reported state-of-the-art organics separation membranes. As demonstrated in Fig. 4g, the membranes prepared by emerging COFs (marked as blue pentagon) present preponderant separation performance but face a trade-off between permeance and rejection. Our photo-tailoring strategy can create heterocrystalline COMs with both high-crystalline regions and low-crystalline regions, allowing for fast and precise organics separation by eliminating non-selective intercrystalline defects. The as-prepared heterocrystalline COM exhibits stable and up to 44-times higher solvent permeance than previously reported COMs with similar rejection. We further evaluate the separation performance of $DP_{2h}COM$ in cross-flow mode for higher concentration dye solution to investigate its potential in practical application circumstances[39]. Our $DP_{2h}COM$ can withstand continuous

cross-flow shear forces, display high rejection to 500 ppm dye feed solution, and exhibit stable separation performance over 48-hour operation (Supplementary Fig. 18), indicating great potential for large-scale organic molecular separation process.

## Discussion

In summary, we propose a concept of heterocrystalline membrane which comprises high-crystalline regions and low-crystalline regions, thus delicately solving the dilemma between high crystallinity and easy fabrication of defect-free membrane. The preparation of heterocrystalline COM is via a two-step procedure, where a high-crystalline COF membrane forms in the first dark reaction step and the low-crystalline regions form in the second photo reaction step. By tuning the photo reaction time, the low-crystalline regions can tightly and flexibly link the high-crystalline regions to acquire the defect-free COMs with ultrahigh porosity. Accordingly, the resulting COM displays sharp molecular sieving properties with remarkable organic solvents permeance up to 44-times higher than the state-of-the-art membranes. We envisage that our strategy of using photo reaction to tailor the crystallinity of COF membranes may enlighten the manufacture of other crystalline polymer materials, and particularly the concept of heterocrystalline membrane will greatly enrich the design of heterostructure membranes for organics separations and other precise separations.

## Methods

**Preparation of DCOM by dark reaction and PCOM by photo reaction.** The DCOMs were prepared by interfacial polymerization. In a typical process, stock solution of Tp was prepared in dichloromethane at a concentration of 0.20 mmol L$^{-1}$. Stock solutions of Bpy and p-Toluenesulfonic acid (PTSA) were prepared in water/acetonitrile 70:30 (vol/vol) solution at a concentration of 0.30 mmol L$^{-1}$ and 0.60 mmol L$^{-1}$. The interfacial polymerization of membrane was conducted in a beaker. First, 100 mL of Tp stock solution was poured into the beaker followed by pouring 60 mL of water/acetonitrile 70:30 (vol/vol) solution as a spacer layer. After that, 100 mL of Bpy and PTSA solution were slowly and evenly added on the top of the spacer solution over 30 min controlled by an autosampler. The reaction mixture was left in a dark box at ambient temperature (25 ± 5 °C) for 96 h. The DCOMs were then formed at the immiscible interface, collected by removing the top aqueous solution, washed with dimethylacetamide (DMAc), acetonitrile, methanol, and acetone in sequence for 48 h to remove the residual monomers and small aggregates/particles. Afterward, they were transferred to substrates and heat-treated in an air oven (90 °C, 48 h).

PCOMs were prepared using the same method of DCOMs but formed under a 300 W Xe lamp (MC-301, Merry Change Technology Co. Ltd., China) with controlled irradiation intensity and equipped with a 200–400 nm UVREF filter.

**Preparation of DPCOM by sequential dark reaction and photo reaction.** The DCOMs were prepared by interfacial polymerization. In a typical process, stock solution of Tp was prepared in dichloromethane at a concentration of 0.20 mmol L$^{-1}$. Stock solutions of Bpy and p-Toluenesulfonic acid (PTSA) were prepared in water/acetonitrile 70:30 (vol/vol) solution at a concentration of 0.30 and 0.60 mmol L$^{-1}$. The interfacial polymerization of membrane was conducted in a beaker. First, 100 mL of Tp stock solution was poured into the beaker followed by pouring 60 mL of water/acetonitrile 70:30 (vol/vol) solution as a spacer layer. After that, 100 mL of Bpy and PTSA solution were slowly and evenly added on the top of the spacer solution over 30 min controlled by an autosampler. The reaction mixture was left in a dark box at ambient temperature (25 ± 5 °C) for 96 h. Then the photo irradiation is in-situ employed on the reaction mixture with intensity of 9.0 mW cm$^{-2}$. After irradiating of 0–2 h, the DPCOMs were collected, washed with solvents, transferred to substrates, and heat-treated in an air oven using the same method of DCOMs.

## Data availability

All data supporting the findings of this study are available within the article and the Supplementary Information file, or available from the corresponding authors upon request. Source data are provided with this paper.

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

## Acknowledgements

The authors gratefully acknowledge financial support from National Natural Science Foundation of China (21878215) and Key Research and Development Program of Zhejiang Province (2021C03173).

## Author contributions

H.W., Z.J., and J.Y. conceived the idea and designed the research. J.Y. and X.Y. carried out the experiment. J.S. and Z.X. performed the permeation measurement. R.L. and Y.L. provided constructive suggestions for results and discussion. L.C., R.Z., N.K., and M.L. helped to revise the paper. All authors participated in the discussion. H.W., Z.J., J.Y., and X.Y. co-wrote the paper.

## Competing interests

The authors declare no competing interests.
