## [Peer Review File · Nature Communications]

Photo-tailored heterocrystalline covalent organic framework membranes for organics separationReviewers' Comments:

Reviewer #1:

Remarks to the Author:

In this work, the authors propose a concept of heterocrystalline membrane, using covalent organic framework membrane (COM) as model membrane to solve the dilemma between high crystallinity and easy fabrication of defect-free COM. The heterocrystalline membrane is formed using a two-stage process and photochemistry to generate low-crystalline COF regions that act to block the defect in a high-crystalline COF system. The resulting membranes are utilized for the separation of organic solvents and achieve the ever-reported highest selectivity and solvent permeances. This work is of interest for the broad community of COF material science, membrane science, and organics separation. Therefore, I would like to suggest accepting this paper after addressing the following minor issues.

1. The authors claim that this photo-tailored crystallinity strategy is practical to Schiff-base COM with a very nice demonstration. I wonder whether the photo irradiation will influence the crystallinity of other kinds of Schiff-base COFs.
2. The authors evaluate the separation performance at a concentration of 50 ppm. Since the organic concentrations in the industry are usually higher than 50 ppm, even ten-fold higher in some cases, so how about the separation performance of the membrane for higher concentration organic solutions?
3. As shown in Fig. 3d, the authors use the full width at half maximum of diffraction peak to explore whether photo irradiation influences the crystalline structure of the high-crystalline regions. Why not use the intensity of the diffraction peak?
4. The intercrystalline defects severely hinder the fabrication of defect-free COM. To cross this hurdle, a common strategy is to increase membrane thickness usually from nanometer to micrometer scale. Have the authors evaluated the separation performance of a thicker DCOM (COM fabricated in the first stage, i.e., COM fabricated in dark reaction) in their study? And particularly is the separation performance comparable with DP2hCOM?
5. Recently, organic solvent nanofiltration has attracted intense interest from membrane science. What is the major distinction between water treatment membranes and organic solvent treatment membranes?

Reviewer #2:

Remarks to the Author:

In this manuscript, a heterocrystalline COF membrane comprising high-crystalline regions and low-crystalline regions was prepared through a two-step procedure based on sequential Schiff-base reactions. The bond linkage can tautomerize under photo irradiation, form low-crystalline regions, and link high-crystalline regions to obtain a defect-free membrane. This strategy delicately solves the dilemma between high crystallinity and easy fabrication of defect-free membrane, providing a new design prototype to process crystalline polymer materials. From this point of view, this work is highly innovative. The experiments are thorough and the manuscript is clearly written. Therefore, I recommend publication after a minor revision. There are a few comments for authors to consider,

1. As shown in Figure S6, the defects on the DCOM gradually decrease with the prolongation of the synthesis time. If the defects of DCOM can be eliminated via prolonging the synthesis time without photo-irradiation?
2. As shown in Figure 3d, the intensity of the (100) diffraction peak of DPCOM decreases after photo-irradiation, while the full width at half maximum of the peak remains unchanged. Detailed discussion should be provided.
3. Have you tried to synthesize the DPCOM with longer photo reaction time (>2h)? How is the performance?

4. Chemical structure of COF membrane should be fully characterized, such as NMR measurement.
5. What is the advance of the proposed strategy compared to other ways of compensating for defects, such as secondary interfacial polymerization? The authors should comment on that.
6. Can you please explain why the initial ethanol contact angle of the different COMs is similar, but other initial solvent (acetonitrile, ethyl acetate, methanol et al.) contact angle is different?
7. The COF film is physically transferred onto the support. I assume that delamination/detach of this COF layer may occur, the authors should comment on the adherence of the COF layer to the support.
8. An important issue is the stability of the COF materials. Did the authors study the effect of water and different pH on their COF membranes?

Reviewer #3:

Remarks to the Author:

In this manuscript, the authors propose a photo-tailoring strategy to fabricate the heterocrystalline COF membranes (COM) with both high-crystalline regions (dark reaction) and low-crystalline regions (photo reaction). It was demonstrated that post-introduction of low-crystalline regions via photo-induced reaction can repair the defects from high-crystalline regions, thereby promoting co-existence of both regions. The resulting COM displays sharp molecular sieving properties with remarkable organic solvents permeance up to 44-times higher than the state-of-the-art membranes. The manuscript is well organized and presented clearly. These findings are interesting and provide new insights in the design of heterocrystalline COM. Considering the overall high quality of this work, I recommend its acceptance by Nature Communication after some minor revisions. The specific comments are as follows.

-The synthesis of imine-linked COF membranes via interfacial polymerization is affected by many factors such as monomer concentration, reaction time, catalysis, etc. While the authors control the same conditions for COM synthesis, it is better to present the key details such as concentration and dark reaction time in the manuscript. In addition, dark reaction time is of importance for synthesis of high-quality heterocrystalline COF. It is suggested to demonstrate such point in the manuscript.

-The concentration of Tp and Bpy is separately fixed at 0.02 mmol and 0.03 mmol for the synthesis. How did the authors make sure that such concentration is optimum parameter for COM performance based on two-step reaction?

-In Figure 3a, the insert cross-sectional images are too small to read; please move them to SI or increase the size of these figures.

-During a dark reaction process, the enol-imine linkage would tautomerize irreversibly to stable keto-enamine form. Which curve is the spectrum of DCOM in Figure S2? Why is the intensity of C=C band very weak? If possible, XPS analysis is required to verify the presence of keto-enamine-linked COM.

-The increase of an irradiation intensity lead to a marked increase of C=C band compared to dark reaction. Please provide a detailed explanation.

-It was found that more rapid reaction rate corresponds to a marked increased of C=C band and more low-crystalline regions. Are there any links between them?

-Is that possible to reverse the sequence of dark reaction and photo reaction for tuning the crystallinity and performance of COMs.

-I am curious about the permeance of n-hexane for DP2hCOM.

-Minor corrections. Keep the format of chemicals purity consistent, such as 98.0%, and 98%

NCOMMS-22-02553-T

TITLE: Photo-tailored heterocrystalline covalent organic framework membranes for organics separation

A detailed response to the reviewers' comments

Reviewers Comments:

Reviewer #1 (Remarks to the Author):

In this work, the authors propose a concept of heterocrystalline membrane, using covalent organic framework membrane (COM) as model membrane to solve the dilemma between high crystallinity and easy fabrication of defect-free COM. The heterocrystalline membrane is formed using a two-stage process and photochemistry to generate low-crystalline COF regions that act to block the defect in a high-crystalline COF system. The resulting membranes are utilized for the separation of organic solvents and achieve the ever-reported highest selectivity and solvent permeances. This work is of interest for the broad community of COF material science, membrane science, and organics separation. Therefore, I would like to suggest accepting this paper after addressing the following minor issues.

Reply:

Thank the reviewer for the highly positive remarks and strong support of publication of this work.

1. The authors claim that this photo-tailored crystallinity strategy is practical to Schiff-base COM with a very nice demonstration. I wonder whether the photo irradiation will influence the crystallinity of other kinds of Schiff-base COFs.

Reply:

Based on the reviewer's valuable guidance, we further prepared another two kinds of Schiff-base COMs (Tp-Tta and Tp-Azo) under either dark condition or photo-condition

with irradiation intensity of 9.0 mW/cm^{-2} . The crystallinity of COMs was evaluated by X-ray diffractometer (XRD). As illustrated in Fig. R1, the Tp-Tta and Tp-Azo COM formed under dark condition show an intense and sharp peak at $\sim 5.8^\circ$ and $\sim 3.2^\circ$, respectively, corresponding to reflections from the (100) lattice plane. By contrast, the Tp-Tta COM and Tp-Azo COM formed under photo-condition display a substantially weaker and wider (100) diffraction peak, indicating the pronounced influence of photo-irradiation on the COM crystallization. The relevant discussions have been added in the revised manuscript and revised supplementary information.

Fig. R1 XRD patterns of the (a) Tp-Tta COM and (b) Tp-Azo COM formed under dark condition (blue) and photo condition (yellow).

The manuscript was revised as follows:

- Section of photo-tailored reactive-crystallization of Schiff-base COM, Paragraph 2

“These findings demonstrate a simple and effective strategy for tailoring the crystalline structure of Tp-Byp COM. To evaluate the generality of this strategy, we further prepared two kinds of Schiff-base COM, Tp-Tta and Tp-Azo. It has been found that both the Tp-Tta and Tp-Azo COM formed by photo reaction exhibit notably less crystallinity than those formed by dark reaction (Supplementary Fig. 7). This strategy offers the possibility to tailor heterocrystalline COM by controlling the dark and photo

reactions during membrane formation.”

The supplementary information was revised as follows:

- Section 2, Supplementary Figure 7

Supplementary Figure 7. XRD patterns of the (a) Tp-Tta COM and (b) Tp-Azo COM formed under dark condition (blue) and photo condition (yellow).

2. The authors evaluate the separation performance at a concentration of 50 ppm. Since the organic concentrations in the industry are usually higher than 50 ppm, even ten-fold higher in some cases, so how about the separation performance of the membrane for higher concentration organic solutions?

Reply:

Thanks for the reviewer’s valuable guidance. We further evaluated the separation performance of the DP_{2h}COM for dye solutions with concentration ranging from 20 ppm to 500 ppm. As shown in Fig. R2, the DP_{2h}COM maintained high dye rejection (>95%) under varied feed concentrations. Additionally, the rejection slightly increases from 95% to 97% when dye concentration increases from 50 ppm to 500 ppm due to the larger dye clusters arising from the more favorable dye aggregation at higher dye concentrations^{1,2}. The relevant discussions have been added in the revised manuscript and revised supplementary information.

Fig. R2 The rejection efficiency of the DP_{2h}COM for Evans blue solution with varied concentrations.

The manuscript was revised as follows:

- Section of organics separation performance of DPCOMs, Paragraph 3

“We further evaluate the separation performance of DP_{2h}COM in cross-flow mode for higher concentration dye solution to investigate its potential in practical application circumstances³⁸. Our DP_{2h}COM can withstand continuous cross-flow shear forces, display high rejection to 500 ppm dye feed solution, and exhibit stable separation performance over 48-hour operation (Supplementary Fig. 18), indicating great potential for large-scale organic molecular separation process.”

The supplementary information was revised as follows:

- Section 1.5, Paragraph 2

“The dyes were dissolved in ethanol at a concentration of 50 ppm (20 ppm for EB) and in 99% (v/v) ethanol-water at a concentration higher than 50 ppm.”

- Section 2, Supplementary Figure 18

Supplementary Figure 18. a, Schematic representation of the cross-flow unit. **b**, The rejection efficiency of the DP_{2h}COM for Evans blue solution with varied concentrations. **c**, long-term operating stability of DP_{2h}COM evaluated by cross-flow unit. The inset digital photograph is the feed and permeate solution, respectively.

3. As shown in Fig. 3d, the authors use the full width at half maximum of diffraction peak to explore whether photo irradiation influences the crystalline structure of the high-crystalline regions. Why not use the intensity of the diffraction peak?

Reply:

Thanks for the reviewer's valuable comments. The full width at half maximum of the diffraction peak can reflect the size of the crystalline domains based on Scherrer equation³.

$$\tau = \frac{K\lambda}{\beta \cos \theta} \quad (1)$$

β is the line broadening at half of the maximum intensity (FWHM), after subtracting the instrumental line broadening, in radians. τ is the size of the ordered (crystalline) domains; K is a dimensionless shape factor, with a value close to unity. The shape factor varies with the actual shape of the crystallite. Herein we have considered the shape factor as unity for the ease of calculation. λ is the X-ray wavelength which has the value 1.5418 Å; θ is the Bragg angle (in degrees). Therefore, we use full width at half maximum of diffraction peak to assess the effect of photo irradiation on the high-crystalline regions of COF membranes.

4. The intercrystalline defects severely hinder the fabrication of defect-free COM. To cross this hurdle, a common strategy is to increase membrane thickness usually from nanometer to micrometer scale. Have the authors evaluated the separation performance of a thicker DCOM (COM fabricated in the first stage, i.e., COM fabricated in dark reaction) in their study? And particularly is the separation performance comparable with DP2hCOM?

Reply:

Thanks for the reviewer's valuable guidance. We further fabricated a thicker DCOM (120h) with thickness of ~130 nm by prolonging the dark reaction time to 120 h. As illustrated in Fig. R3, although the rejection could be improved, the fabricated membrane showed only moderate ethanol permeance of about 25 L·m⁻²·h⁻¹·bar⁻¹, which was 69% less than that of DP_{2h}COM, arising from the significantly increased mass transfer resistance due to the increased thickness.

Fig. R3 **a-c**, Cross-sectional SEM images of (a) DCOM (96h), (b) DCOM (120h) and (c) DP_{2h}COM. **d**, Pure ethanol permeance and Evans blue rejection of DCOM (96 h), DCOM (120 h) and DP_{2h}COM.

5. Recently, organic solvent nanofiltration has attracted intense interest from membrane science. What is the major distinction between water treatment membranes and organic solvent treatment membranes?

Reply:

The major distinctions between water treatment membranes and organic solvent treatment membranes are summarized in Table 1.

Table 1 Examples of key applications of nanofiltration membranes in water treatment and organic solvent treatment, and analogies between the most critical challenges in each domain.

	Water treatment membrane ⁴⁻⁷	Organic solvent treatment membrane ^{4,8-11}
Key applications	a. Seawater and brackish groundwater desalination b. Waste water recovery	a. Catalyst recovery b. Pharmaceutical enrichment and purification c. Aromatics/alkanes separation

		d. Waste organic solvent (e.g. apolar, polar protic, and polar aprotic solvent) recovery
Overarching challenges	a. Improving selectivity in salt/water, dye/water, divalent/monovalent salt ion, and salt/dye. b. Overcoming permeability/selectivity trade-offs c. Reducing fouling by precipitated salts and biofilms d. Enhancing acid and alkali resistance to enable complete regeneration of membranes	a. Improving selectivity in organic solute A/organic solute B ($M_{w_{solute}}=200-1000$ Da). b. Overcoming permeability/selectivity trade-offs c. Reducing fouling by high molecular species such as dissolved organic matter d. Minimizing degradation in harsh organic solvents and develop robust regeneration protocols to enhance membrane lifetime

Reviewer #2 (Remarks to the Author):

In this manuscript, a heterocrystalline COF membrane comprising high-crystalline regions and low-crystalline regions was prepared through a two-step procedure based on sequential Schiff-base reactions. The bond linkage can tautomerize under photo irradiation, form low-crystalline regions, and link high-crystalline regions to obtain a defect-free membrane. This strategy delicately solves the dilemma between high crystallinity and easy fabrication of defect-free membrane, providing a new design prototype to process crystalline polymer materials. From this point of view, this work is highly innovative. The experiments are thorough and the manuscript is clearly written. Therefore, I recommend publication after a minor revision. There are a few comments for authors to consider,

Reply:

Thank the reviewer for the highly positive remarks and strong support of publication of this work.

1. As shown in Figure S6, the defects on the DCOM gradually decrease with the prolongation of the synthesis time. If the defects of DCOM can be eliminated via prolonging the synthesis time without photo-irradiation?

Reply:

Thanks for the reviewer's valuable guidance. We investigated the structure and separation performance of DCOMs fabricated in dark reaction for longer time. As shown in Fig. R4, longer fabrication time yielded DCOMs with reduced defects. The Evans blue rejection of DCOM rises from 45% to 93% when fabrication time is increased from 96 to 128 hours (Fig. R5), confirming the decreasing defects in D_{128h}COM. Nonetheless, the Evans blue rejection of D_{128h}COM is not as high as that of DP_{2h}COM (99%), indicating that the low-crystalline regions of DP_{2h}COM generated under photo irradiation are efficient at sealing membrane defects. Additionally, the ethanol permeance of D_{128h}COM

falls dramatically to $32 \text{ L m}^{-2} \text{ h}^{-1} \text{ bar}^{-1}$, 60% less than that of $\text{DP}_{2\text{h}}\text{COM}$, owing to the reduced defects and increased membrane thickness.

Fig. R4 Cross-sectional SEM images of DCOMs prepared with reaction time of (a) 96 h, (b) 112 h and (c) 120 h.

Fig. R5 Pure ethanol permeance and Evans blue rejection of DCOM (96 h), DCOM (120 h) and $\text{DP}_{2\text{h}}\text{COM}$.

2. As shown in Figure 3d, the intensity of the (100) diffraction peak of DPCOM decreases after photo-irradiation, while the full width at half maximum of the peak remains

unchanged. Detailed discussion should be provided.

Reply:

Thanks for the reviewer's valuable comments. The full width at half maximum of the diffraction peak reflects the size of the crystalline domains based on Scherrer equation³, and the intensity of the diffraction peak reflects the relative crystallinity of membrane¹². After photo reaction, the low-crystalline regions of DPCOM forms, while the high-crystalline regions remain unchanged. Thus, the full width at half maximum of the (100) diffraction peak remains unchanged, but its intensity decreases because of the increased proportion of low-crystalline regions (Fig. R6).

Fig. R6 XRD patterns of DCOM, DPCOM and PCOM (9.0 mW cm^{-2}).

3. Have you tried to synthesize the DPCOM with longer photo reaction time (>2h)? How is the performance?

Reply:

Thanks for the reviewer's valuable comments. We further investigated the separation performance of DPCOM fabricated with longer photo reaction time. As shown in Fig. R7, when the photo reaction time is extended to 2.5 hours, the pure ethanol permeance further decreases to $47 \text{ L m}^{-2} \text{ h}^{-1} \text{ bar}^{-1}$, while the Evans blue rejection remains at 99%.

Fig. R7 Pure ethanol permeance and Evans blue rejection of COMs. The PCOM here is fabricated under 9.0-mW cm⁻² irradiation

4. Chemical structure of COF membrane should be fully characterized, such as NMR measurement.

Reply:

Based on the reviewer's valuable guidance, we further characterized the chemical structure of COF membrane by ¹³C cross-polarization magic angle spinning (CP-MAS) solid-state nuclear magnetic resonance (NMR) spectroscopy. As illustrated in Fig. R8, the spectrum of DCOM exhibits resonances at 151.0 ppm and 107.6 ppm, corresponding to the enamine carbon and the α -enamine carbon, respectively, which is consistent with the reported β -ketoenamine-linked COF membrane^{13,14}. The spectrum of PCOM shows the same resonances as that of DCOM, while the peaks are wider and less resolved, suggesting low structural orderness¹⁵. The relevant discussions have been added in the revised manuscript and revised supplementary information.

Fig. R8 ^{13}C solid-state NMR spectra of DCOM and PCOM (9.0 mW cm^{-2}). Carbon atoms responsible for the NMR resonances are labelled a-c, Py.

The manuscript was revised as follows:

- Section of photo-tailored reactive-crystallization of Schiff-base COM, Paragraph 1

“Fourier transform infrared (FTIR), solid-state ^{13}C nuclear magnetic resonance (NMR) and X-ray photoelectron spectrometer (XPS) spectra confirm the formation of the keto-enamine-linked COM by dark reaction (DCOM), as indicated by the C=C stretching band at ca. 1566 cm^{-1} (Supplementary Fig. 4), enamine carbon resonance at 151.0 ppm (Supplementary Fig. 5), and secondary amine (=C-NH) with binding energy of 399.8 eV (Supplementary Fig. 6).”

- Section of photo-tailored reactive-crystallization of Schiff-base COM, Paragraph 2

“NMR spectra reveal that the resonance peaks of PCOM are wider and less resolved than those of DCOM, confirming the poor development of crystalline structure in PCOM (Supplementary Fig. 5)³².”

The supplementary information was revised as follows:

- Section 1.4

1.4.4 Nuclear magnetic resonance spectroscopy (NMR)

Solid-state ^{13}C cross-polarization magic angle spinning (CP-MAS) NMR spectra were performed on a Varian infinity plus 300 NMR spectrometer under 12 kHz spinning rate.

The samples were dried in hot air oven at 60 °C for 12 h and diced into small pieces before NMR analysis.

- Section 2, Supplementary Figure 5

Supplementary Figure 1. ¹³C solid-state NMR spectra of DCOM and PCOM (9.0 mW cm⁻²). Carbon atoms responsible for the NMR resonances are labelled a-c, Py.

Note: The spectrum of DCOM exhibits resonances at 151.0 ppm and 107.6 ppm, corresponding to the enamine carbon and the α-enamine carbon, respectively, which is consistent with the reported β-ketoenamine-linked COF membrane^{2,3}. The spectrum of PCOM shows the same resonances as DCOM, while the peaks are wider and less resolved, suggesting a lower crystalline structure⁴.

5. What is the advance of the proposed strategy compared to other ways of compensating for defects, such as secondary interfacial polymerization? The authors should comment on that.

Reply:

Intercrystalline defects are prevalent in crystalline membranes as a result of poor crystal intergrowth and can deteriorate the membrane separation selectivity. Currently, several strategies have been developed for the manipulating grain boundary structures and minimizing the effects of intercrystalline defects on the membrane performance. Secondary growth strategy *via* interfacial polymerization^{16,17} and direct shielding strategy

via depositing large molecules^{18,19} are all effective post-remediation methods to block the intercrystalline defects and enhance the separation selectivity after membrane fabrication. However, the challenge in improving the compatibility of first growth crystals with secondary growth crystals, or crystals with large molecules, remains a critical issue. For our heterocrystalline COF membrane prepared using a photo-tailored crystallinity strategy, the chemical compositions of the high-crystalline regions and the low-crystalline regions are the same, and the low-crystalline region with good membrane-formation ability can tightly and flexibly link the rigid high-crystalline region, effectively blocking intercrystalline defects. In addition, this photo-tailored strategy is facile, requiring no sophisticated post-processing procedures.

6. Can you please explain why the initial ethanol contact angle of the different COMs is similar, but other initial solvent (acetonitrile, ethyl acetate, methanol et al.) contact angle is different?

Reply:

Thanks for the reviewer's valuable comments. The initial solvent contact angle reflects the surface wettability as the solvent drop comes in to contact with the membrane surface before spreading and penetration. The as-prepared DCOM, DP_{2h}COM and PCOM all display superior organophilic behavior, the initial contact angles of various solvents are all less than 20° and decline to 0° in a few seconds. The contact angle of acetonitrile, ethyl acetate and methanol decline more rapidly than that of ethanol, because the penetration of the low-viscosity solvents of acetonitrile (3.4×10^{-4} Pa·s), ethyl acetate (4.26×10^{-4} Pa·s) and methanol (5.4×10^{-4} Pa·s) through COMs is faster than that of high-viscosity solvent of ethanol (1.08×10^{-3} Pa·s), which poses a challenge for the contact angle goniometer to record the instantaneous contact angle accurately. Thus, the difference in initial solvent contact angle (acetonitrile, ethyl acetate, methanol et al.) between various COMs is slightly higher than the difference in initial ethanol contact angle.

7. The COF film is physically transferred onto the support. I assume that delamination/detach of this COF layer may occur, the authors should comment on the adherence of the COF layer to the support.

Reply:

In this study, polyethylene terephthalate (PET) microfiltration membrane was chosen as the substrate because of their high flexibility and abundance of ester groups that could form hydrogen bonds with the secondary amine groups and terminal primary amine groups of COF layer (Fig. R9). Additionally, the morphology of the skin layer is another non-negligible factor, that is, the rougher surface with greater surface area usually leads to higher adhesion strength between the skin layer and the substrate²⁰. As shown in Fig. R10, the COF layer generated by interfacial polymerization bears a rough surface on the side towards aqueous phase. When the COF layer is transferred to the substrate, the rough side faces down, providing a much higher surface area for anchoring the substrate. Fig. R11 shows the digital photos of DP_{2h}COM deposited on PET substrate remaining intact after folding. Furthermore, it is well-known that the adhesion property between the substrate and the skin layer is crucial in determining the stable performance of thin film composite membranes. Fig. R12 shows that the DP_{2h}COM deposited on PET substrate is capable of withstanding continuous cross-flow shear forces and maintaining separation performance during 48-hour operation. These results indicate that the COF layer adheres well to the substrate.

Fig. R9 The schematic of the skin-substrate adhesion strength.

Fig. R10 SEM images of the DCOM generated by interfacial polymerization.

Fig. R11 Photos of $DP_{2h}COM$ deposited on PET substrate. It remains intact and undamaged after folding.

Fig. R12 a, Schematic representation of the cross-flow unit. b, long-term operating stability of $DP_{2h}COM$ evaluated by cross-flow unit. The inset digital photograph is the feed and permeate solution, respectively

8. An important issue is the stability of the COF materials. Did the authors study the effect of water and different pH on their COF membranes?

Reply:

Based on the reviewer's valuable guidance, we evaluated the pH stability of COF membrane in both alkaline and acid solutions, aqueous sodium hydroxide (NaOH) solution (pH 13) and hydrochloric acid (HCl) solution (pH 2), respectively. The membranes were immersed in the above alkaline or acid solutions at room temperature. As shown in Fig. R13, the rejection of DP_{2.5h}COM decreases by only 5% after immersion in strong alkaline solution or acid solution for 72 h, exhibiting good stability under extreme pH conditions.

Fig. R13 The rejection of DP_{2.5h}COM after immersion in acid aqueous solution (yellow) and alkaline aqueous solution (blue).

Reviewer #3 (Remarks to the Author):

In this manuscript, the authors propose a photo-tailoring strategy to fabricate the heterocrystalline COF membranes (COM) with both high-crystalline regions (dark reaction) and low-crystalline regions (photo reaction). It was demonstrated that post-introduction of low-crystalline regions via photo-induced reaction can repair the defects from high-crystalline regions, thereby promoting co-existence of both regions. The resulting COM displays sharp molecular sieving properties with remarkable organic solvents permeance up to 44-times higher than the state-of-the-art membranes. The manuscript is well organized and presented clearly. These findings are interesting and provide new insights in the design of heterocrystalline COM. Considering the overall high quality of this work, I recommend its acceptance by Nature Communication after some minor revisions. The specific comments are as follows.

Reply:

Thank the reviewer for the highly positive remarks and strong support of publication of this work.

1. The synthesis of imine-linked COF membranes via interfacial polymerization is affected by many factors such as monomer concentration, reaction time, catalysis, etc. While the authors control the same conditions for COM synthesis, it is better to present the key details such as concentration and dark reaction time in the manuscript. In addition, dark reaction time is of importance for synthesis of high-quality heterocrystalline COFM. It is suggested to demonstrate such point in the manuscript.

Reply:

Before synthesizing heterocrystalline DPCOM, we first optimized the reaction time and monomer concentration to obtain a high-crystalline and ultrathin DCOM. X-ray diffraction (XRD) patterns suggest that the crystallinity of DCOMs rises to a high point with high peak intensity at 96 h and increase slightly with further increases in time (Fig.

R14). Cross-sectional scanning electron microscopy (SEM) indicates the thickness of the DCOMs decreases to ~55 nm as the concentration of Bpy and Tp decreases to 0.30 mmol L⁻¹ and 0.20 mmol L⁻¹, respectively (Fig. R15). Top-view SEM demonstrates the fibre-like crystal assembly morphology of high-crystalline DCOM and the inter-crystal defects are obviously observed when the membrane thickness decreased to below 100 nm. Further reducing the monomer concentration, the membrane thickness remained around 50 nm, but the inter-crystal defects became more severe. Hence, we chose reaction time of 96 h, Bpy concentration of 0.30 mmol L⁻¹ and Tp concentration of 0.20 mmol L⁻¹ as the optimum condition parameters. The relevant discussions have been added in the revised manuscript and revised supplementary information.

Fig. R14 XRD spectra monitoring the change in crystallinity of DCOMs with increasing fabrication time.

Fig. R15 Top-view (left) and cross-sectional (right) SEM images of DCOMs fabricated with varied monomer concentration. The molar ratio of Bpy (diamine) and Tp (trialdehyde) is constant at 3:2.

The manuscript was revised as follows:

- Section of Photo-tailored reactive-crystallization of Schiff-base COM, Paragraph 1

“The optimal reaction time was set at 96 hours, and the Bpy and Tp concentrations were set at 0.30 and 0.20 mmol L⁻¹, respectively (Supplementary Fig. 2, 3).”

The supplementary information was revised as follows:

- Section 2, Supplementary Figure 2

Supplementary Figure 2. XRD spectra monitoring the change in crystallinity of DCOMs with increasing fabrication time.

Note: To obtain a high-crystalline DCOM, we optimized the reaction time. The XRD patterns of DCOMs prepared at different reaction time suggest that the crystallinity of membranes increased with the reaction time and a high crystallinity could be achieved at 96 h and thereafter. Hence, reaction time of 96 h was selected as the optimum reaction time.

▪ Section 2, Supplementary Figure 3

Supplementary Figure 3. Top-view (left) and cross-sectional (right) SEM images of DCOMs fabricated with varied monomer concentration. The molar ratio of Bpy (diamine) and Tp (trialdehyde) is constant at 3:2.

Note: We adjusted the monomer concentration to achieve an ultrathin DCOM. Cross-sectional SEM indicates the thickness of the DCOMs decreases to ~55 nm as the concentrations of Bpy and Tp decrease to 0.30 mmol L⁻¹ and 0.20 mmol L⁻¹, respectively. Top-view SEM demonstrates the fibre-like crystal assembly morphology of high-crystalline DCOM and the inter-crystal defects are obviously observed when the membrane thickness decreased to below 100 nm. Further reducing the monomer

concentration, the membrane thickness remained around 50 nm, but the inter-crystal defects became more severe. Hence, Bpy concentration of 0.30 mmol L⁻¹ and Tp concentration of 0.20 mmol L⁻¹ was selected as the optimum concentration.

2. The concentration of Tp and Bpy is separately fixed at 0.02 mmol and 0.03 mmol for the synthesis. How did the authors make sure that such concentration is optimum parameter for COM performance based on two-step reaction?

Reply:

Thanks for the reviewer's valuable comments. In the reply to question 1 of this reviewer, we have explained why the concentrations of Tp and Bpy are separately fixed at 0.2 mmol L⁻¹ and 0.3 mmol L⁻¹, or 0.02 mmol and 0.03 mmol of amount, respectively.

3. In Figure 3a, the insert cross-sectional images are too small to read; please move them to SI or increase the size of these figures.

Reply:

Based on the reviewer's valuable guidance, the cross-sectional images insert in Figure 3a were moved to SI.

The manuscript was revised as follows:

- Section of preparation and characterizations of DPCOMs, line 123-125

“Moreover, the thickness of the DP_{2h}COM does not increase (~55 nm), suggesting that the low-crystalline regions grow in the intercrystalline defects instead of along the thickness of the membrane (Supplementary Fig. 11).”

- Section of preparation and characterizations of DPCOMs, Fig. 3

Fig. 3 | Structure and morphology characterizations. **a**, Top-view SEM images of COMs deposited on track-etched substrate membranes. The intercrystalline defects are marked by the white circle. Scale bar = 500 nm.

The supplementary information was revised as follows:

- Section 2, Supplementary Figure 11

Supplementary Figure 11. The cross-sectional SEM images inset showing the thickness of COMs. Scale bar = 500 nm.

4. During a dark reaction process, the enol-imine linkage would tautomerize irreversibly to stable keto-enamine form. Which curve is the spectrum of DCOM in Figure S2? Why is the intensity of C=C band very weak? If possible, XPS analysis is required to verify the presence of keto-enamine-linked COM.

Reply:

Thanks for the reviewer's valuable comments. The blue curve named "0 mW cm⁻¹" is the spectrum of DCOM. We have changed the name to "DCOM" to make it clearer.

During the dark reaction process, the enol-imine linkage would tautomerize irreversibly to stable keto-enamine form (CO-C=C-NH) until the initial amorphous network convert to crystalline framework²¹. By contrast, during a photo reaction process, the phototautomerization of the enol-imine linkage would proceed via a low barrier

transition state without the limitation of crystalline structure. Therefore, the keto-enamine linkage (CO-C=C-NH) of DCOM is less than that of PCOM, and the C=C stretching band (1566 cm^{-1}) of keto-enamine linkage is weaker. The related discussion in the manuscript was revised.

Based on the reviewer's valuable guidance, we performed XPS to verify the presence of keto-enamine-linked COM. As shown in Fig. R16, the N 1s spectra of both DCOM and PCOM can be deconvoluted into three types of nitrogen: pyridinic N at 398.6 eV, secondary amine (=C-NH) at 399.8 eV and chemisorbed nitrogen oxide species at 403 eV, respectively²²⁻²⁴. The secondary amine originates from the Schiff base reaction between the aldehyde monomer of Tp and the amine monomer of Bpy, indicating the formation of keto-enamine-linked COM. Besides, the area ratio of secondary amine/pyridinic N in DCOM is less than that in PCOM, suggesting that the keto-enamine linkage in DCOM is less, agreeing with the FTIR analysis. The analysis of XPS was added in the manuscript.

Fig. R16 High-resolution XPS spectra of the N 1s of (a) DCOM and (b) PCOM (formed under photo irradiation of 9.0 mW cm^{-2}).

The manuscript was revised as follows:

- Section of photo-tailored reactive-crystallization of Schiff-base COM, paragraph 1

“The synthetic routes of COMs by either dark reaction or photo reaction were illustrated in Fig. 2a. Initially, precursor trialdehyde (Tp) and diamine (Bpy) would

polymerize into an amorphous network *via* enol-imine linkage²⁵. During dark reaction, the reversible enol-imine linkage breaks and reforms slowly, thus converting the initial amorphous network into the thermodynamically stable crystalline framework as a result of the "error-correcting" process²⁶. Then, the enol-imine linkage tautomerizes irreversibly to stable keto-enamine form because the basicity of three imine nitrogens (C=N) dominates over the aromaticity of the central benzene ring^{27,28}. Fourier transform infrared (FTIR), Solid-state nuclear magnetic resonance (NMR) and X-ray photoelectron spectrometer (XPS) spectra confirm the formation of the keto-enamine-linked COM by dark reaction (DCOM), as indicated by the C=C stretching band at ca. 1566 cm⁻¹ (Supplementary Fig. 4), enamine carbon resonance at 151.0 ppm (Supplementary Fig. 5), and secondary amine (=C-NH) with binding energy of 399.8 eV (Supplementary Fig. 6)."

- Section of photo-tailored reactive-crystallization of Schiff-base COM, paragraph 2

"Besides, the FTIR spectra demonstrate that the C=C stretching bands in the keto-enamine linkage of PCOMs was more intense than that of DCOM (Supplementary Fig. 4), ascribing from the small energy barriers of enol-keto phototautomerization."

The supplementary information was revised as follows:

- Section 1.4

1.4.5 X-ray photoelectron spectrometer (XPS)

XPS spectra were performed using a K-Alpha+ spectrometer (ThermoFisher Scientific) and an Al-Ka x-ray source under high vacuum (5×10^{-8} Pa). All binding energies were calibrated using C1s peak from the adventitious carbon at 284.80 eV.

- Section 2, Supplementary Figure 10

Supplementary Figure 10. FT-IR spectra monitoring the change in COMs chemical structure with increasing the irradiation intensity.

▪ Section 2, Supplementary Figure 6

Supplementary Figure 6. High-resolution XPS spectra of the N 1s of (a) DCOM and (b) PCOM (9.0 mW cm⁻²).

5. The increase of an irradiation intensity leads to a marked increase of C=C band compared to dark reaction. Please provide a detailed explanation.

Reply:

Thanks for the reviewer's valuable comments. The increase of the irradiation intensity leads to an increased number of absorbed photons per unit time²⁵, thus promoting the phototautomerization of the enol-imine linkage, resulting in PCOMs with more keto-enamine form linkage (CO-C=C-NH). The relevant discussions have been added in the revised supplementary information.

The supplementary information was revised as follows:

- Section 2, Supplementary Figure 10

Supplementary Figure 10. FT-IR spectra monitoring the change in COMs chemical structure with increasing the irradiation intensity.

Note: With the increase of irradiation intensity, the C=C stretching band in the keto-enamine linkage of PCOMs became more intense. The increase of an irradiation intensity leads to an increased number of absorbed photons per unit time¹, thus promoting the phototautomerization of the enol-imine linkage, resulting in PCOMs with more keto-enamine linkage.

6. It was found that more rapid reaction rate corresponds to a marked increase of C=C band and more low-crystalline regions. Are there any links between them?

Reply:

Thanks for the reviewer's valuable comments. Rapid phototautomeric reaction can promote the tautomerization of enol-imine linkage, resulting in a marked increase in the keto-enamine form linkage (CO-C=C-NH) which contains C=C bond. The reversible enol-imine linkage could break and reform during the formation of COMs, allowing the initial mismatched amorphous structure to convert into a high-crystalline structure²⁶. Under photo condition, rapid phototautomeric reaction leads in less reversible enol-imine linkage and more irreversible keto-enamine form linkage containing C=C bond, preventing the "error-correcting" of structure and generating more low-crystalline regions.

7. Is that possible to reverse the sequence of dark reaction and photo reaction for tuning the crystallinity and performance of COMs.

Reply:

Thanks for the reviewer's valuable comments. We reversed the reaction sequence of DP_{2h}COM and prepared P_{2h}DCOM by 2-hour photo reaction first and dark reaction after. The crystallinity of COMs was measured by X-ray diffractometry (XRD). As shown in Fig. R17a, P_{2h}DCOM displays a weak and wide (100) diffraction peak with full width at half maximum (FWHM) of 1.32°, indicating that its crystallinity is inferior to that of DP_{2h}DCOM. We assume that this phenomenon is because the initially formed low-crystalline crystals by photo reaction affect the subsequent morphology evolution of P_{2h}DCOM²⁷⁻²⁹. The separation performance of P_{2h}DCOMs is shown in Fig. R17b, which demonstrates a high ethanol permeance of 148 L m⁻² h⁻¹ bar⁻¹ but only a moderate dye rejection of 95%. The rejection is less than that of DP_{2h}COM (99%). It is demonstrated that conducting dark reaction first and then photo reaction is more effective in eliminating the non-selective intercrystalline defects in COMs than the reverse procedure, thus leading to a higher rejection.

Fig. R17 **a**, XRD patterns of DP_{2h}COM and P_{2h}DCOM. **b**, Pure ethanol permeance and Evans Blue rejection of DP_{2h}COM and P_{2h}DCOM.

8. I am curious about the permeance of n-hexane for DP_{2h}COM.

Reply:

Based on reviewer's valuable guidance, we further evaluated the performance of n-hexane for DP_{2h}COM. As shown in Fig. R18, n-hexane, with the lowest viscosity of 2.97×10^{-4} Pa·s, gives the highest permeance $343 \text{ L m}^{-2} \text{ h}^{-1} \text{ bar}^{-1}$, obeying the viscous flow mechanism fairly well^{30,31}. The permeance of n-hexane for DP_{2h}COM has been added in the revised manuscript and revised supplementary information.

Fig. R18 Permeance of pure organic solvents through the DP_{2h}COM as a function of their inverse viscosity.

The manuscript was revised as follows:

- Section of organics separation performance of DPCOMs, Fig. 4

Fig. 4 | Organics separation performance. d, Permeance of pure organic solvents through the DP_{2h}COM as a function of their inverse viscosity.

▪ Section of organics separation performance of DPCOMs, Paragraph 2

“...and then we further evaluated its permeation properties for different types of organic solvents including apolar (toluene, n-hexane), polar protic (methanol, isopropanol, butanol)...”

“The solvent permeance of DP_{2h}COM is found to be linearly proportional to the inverse of solvent viscosity (η^{-1}). n-Hexane, with a viscosity of $2.97 \times 10^{-4} Pa \cdot s$, gives the highest permeance of $343 L m^{-2} h^{-1} bar^{-1}$ (Supplementary Table 4). Methanol, the most used model solvent, with a small viscosity of $5.4 \times 10^{-4} Pa \cdot s$, also gives high permeances of $264 L m^{-2} h^{-1} bar^{-1}$.”

The supplementary information was revised as follows:

▪ Section 2, Supplementary Table 4

Supplementary Table 4. Solvent viscosity of the organic solvents used in this study

Solvent	Viscosity at 25 °C* (mPa·s)
n-Hexane	0.297
Acetonitrile	0.34
Ethyl acetate	0.426
Tetrahydrofuran	0.457
Acetone	0.31
N,N-Dimethylformamide	0.816
Methanol	0.54
Ethanol	1.08
Isopropanol	2.058
Butanol	2.63
Toluene	0.55

Note: *These data are taken from reference²².

9. Minor corrections. Keep the format of chemicals purity consistent, such as 98.0%, and 98%.

Reply:

Thanks for the reviewer’s valuable guidance. We have carefully checked and corrected the format of chemicals purity in the manuscript and supplementary information.

The supplementary information was revised as follows:

- Section 1.1.2 Paragraph 1

“1,3,5-triformylphloroglucinol (Tp, 98%) was purchased from Yanshen Development Co., Ltd (China). 2,2'-bipyridine-5,5'-diamine (Bpy, 98%) was purchased from Tensus Biotechnology Development Co., Ltd (China).”

In addition, we have carefully and thoroughly checked the manuscript and improved as follows:

Manuscript:

- Section of preparation and characterization of DPCOMs, Paragraph 2

“The heterocrystalline COM, denoted as DPCOM, was fabricated by dark reaction

first in the same way as DCOM, followed by photo reaction under 9.0-mW cm⁻² irradiation (Fig.1).”

- Section of organics separation performance of DPCOMs, Fig. 4

“The PCOM here is fabricated under 9.0-mW cm⁻² irradiation.”

References

- 1 Ye, W. *et al.* Advanced desalination of dye/NaCl mixtures by a loose nanofiltration membrane for digital ink-jet printing. *Sep. Purif. Technol.* **197**, 27-35 (2018).
- 2 Koyuncu, I. Reactive dye removal in dye/salt mixtures by nanofiltration membranes containing vinylsulphone dyes: effects of feed concentration and cross flow velocity. *Desalination* **143**, 243-253 (2002).
- 3 Karak, S., Kumar, S., Pachfule, P. & Banerjee, R. Porosity prediction through hydrogen bonding in covalent organic frameworks. *J. Am. Chem. Soc.* **140**, 5138-5145 (2018).
- 4 Lively, R. P. & Sholl, D. S. From water to organics in membrane separations. *Nat. Mater.* **16**, 276-279 (2017).
- 5 Park, H. B. *et al.* Maximizing the right stuff: The trade-off between membrane permeability and selectivity. *Science* **356**, eaab0530 (2017).
- 6 Lu, X. & Elimelech, M. Fabrication of desalination membranes by interfacial polymerization: History, current efforts, and future directions. *Chem. Soc. Rev.* **50**, 6290-6307 (2021).
- 7 Werber, J. R., Osuji, C. O. & Elimelech, M. Materials for next-generation desalination and water purification membranes. *Nat. Rev. Mater.* **1**, 1-15 (2016).
- 8 Marchetti, P., Solomon, M. F. J., Szekely, G. & Livingston, A. G. Molecular

- separation with organic solvent nanofiltration: A critical review. *Chem. Rev.* **114**, 10735-10806 (2014).
- 9 Marchetti, P., Peeva, L. & Livingston, A. The selectivity challenge in organic solvent nanofiltration: Membrane and process solutions. *Annu. Rev. Chem. Biomol. Eng.* **8**, 473-497 (2017).
 - 10 Hai Anh Le, P., Blanford, C. F. & Szekely, G. Reporting the unreported: The reliability and comparability of the literature on organic solvent nanofiltration. *Green Chem.* **22**, 3397-3409 (2020).
 - 11 Vandezande, P., Gevers, L. E. M. & Vankelecom, I. F. J. Solvent resistant nanofiltration: Separating on a molecular level. *Chem. Soc. Rev.* **37**, 365-405 (2008).
 - 12 Peng, S. *et al.* Control of surface barriers in mass transfer to modulate methanol-to-olefins reaction over SAPO-34 zeolites. *Angew. Chem. Int. Ed. Engl.* **59**, 21945-21948 (2020).
 - 13 DeBlase, C. R. *et al.* β -Ketoenamine-linked covalent organic frameworks capable of pseudocapacitive energy storage. *J. Am. Chem. Soc.* **135**, 16821-16824 (2013).
 - 14 Wang, X. *et al.* Assembling covalent organic framework membranes with superior ion exchange capacity. *Nat. Commun.* **13**, 1020 (2022).
 - 15 Zhang, W. *et al.* Reconstructed covalent organic frameworks. *Nature* **604**, 72-79 (2022).
 - 16 Wang, R., Wei, M. & Wang, Y. Secondary growth of covalent organic frameworks (COFs) on porous substrates for fast desalination. *J. Membr. Sci.* **604**, 118090 (2020).
 - 17 Zheng, Y. *et al.* 2D nanosheets seeding layer modulated covalent organic framework membranes for efficient desalination. *Desalination* **532**, 115753 (2022).

- 18 Huang, A. *et al.* Bicontinuous zeolitic imidazolate framework ZIF-8@GO membrane with enhanced hydrogen selectivity. *J. Am. Chem. Soc.* **136**, 14686-14689 (2014).
- 19 Hong, S. *et al.* Healing of microdefects in SSZ-13 membranes via filling with dye molecules and its effect on dry and wet CO₂ separations. *Chem. Mater.* **30**, 3346-3358 (2018).
- 20 Shi, Q. *et al.* Poly(p-phenylene terephthamide) embedded in a polysulfone as the substrate for improving compaction resistance and adhesion of a thin film composite polyamide membrane. *J. Mater. Chem. A* **5**, 13610-13624 (2017).
- 21 Kandambeth, S. *et al.* Construction of crystalline 2D covalent organic frameworks with remarkable chemical (acid/base) stability via a combined reversible and irreversible route. *J. Am. Chem. Soc.* **134**, 19524-19527 (2012).
- 22 Hota, M. K. *et al.* Electrochemical thin-film transistors using covalent organic framework channel. *Adv. Funct. Mater.*, 2201120 (2022).
- 23 Kumar, S. *et al.* Norbornane-based covalent organic frameworks for gas separation. *Nanoscale* **14**, 2475-2481 (2022).
- 24 Ha, Y. *et al.* Atomically dispersed co-Pyridinic N-C for superior oxygen reduction reaction. *Adv. Energy Mater.* **10**, 2002592 (2020).
- 25 Amezaga-Madrid, P., Nevarez-Moorillon, G., Orrantia-Borunda, E. & Miki-Yoshida, M. Photoinduced bactericidal activity against *Pseudomonas aeruginosa* by TiO₂ based thin films. *FEMS Microbiol. Lett.* **211**, 183-188 (2002).
- 26 Rowan, S. J. *et al.* Dynamic covalent chemistry. *Angew. Chem. Int. Ed.* **41**, 898-952 (2002).

- 27 Yuan, Y. F. *et al.* Electrodeposited growth habit and growth mechanism of ZnO as anode material of secondary alkaline Zn battery. *J. Electrochem. Soc.* **153**, A1719-A1723 (2006).
- 28 Wang, L. *et al.* Study on the morphology-controlled synthesis of MnCO₃ materials and their enhanced electrochemical performance for lithium ion batteries. *Crystengcomm* **18**, 8072-8079 (2016).
- 29 Guerra-Nunez, C. *et al.* Morphology and crystallinity control of ultrathin TiO₂ layers deposited on carbon nanotubes by temperature-step atomic layer deposition. *Nanoscale* **7**, 10622-10633 (2015).
- 30 Yang, Q. *et al.* Ultrathin graphene-based membrane with precise molecular sieving and ultrafast solvent permeation. *Nat. Mater.* **16**, 1198-1202 (2017).
- 31 Dobrak, A. *et al.* Solvent flux behavior and rejection characteristics of hydrophilic and hydrophobic mesoporous and microporous TiO₂ and ZrO₂ membranes. *J. Membr. Sci.* **346**, 344-352 (2010).

Reviewers' Comments:

Reviewer #1:

Remarks to the Author:

The authors have addressed all my concerns and this paper can be accepted as its current form now.

Reviewer #2:

Remarks to the Author:

The authors have properly answered all the questions. I suggest that the manuscript can be accepted as it is now.

Reviewer #3:

Remarks to the Author:

The authors have well addressed the comments and concerns proposed by the reviewers, and provided a detailed discussion on the questions or issues. The revised paper is in a very high quality. Thus, I would like to recommend its acceptance by Nature Communications in its current form.

NCOMMS-22-02553B

TITLE: Photo-tailored heterocrystalline covalent organic framework membranes for organics separation

A detailed response to the reviewers' comments

Reviewer #1 (Remarks to the Author):

The authors have addressed all my concerns and this paper can be accepted as its current form now.

Reply:

Thank the reviewer for the highly positive remarks and support of publication of this work.

Reviewer #2 (Remarks to the Author):

The authors have properly answered all the questions. I suggest that the manuscript can be accepted as it is now.

Reply:

Thank the reviewer for the highly positive remarks and support of publication of this work.

Reviewer #3 (Remarks to the Author):

The authors have well addressed the comments and concerns proposed by the reviewers, and provided a detailed discussion on the questions or issues. The revised paper is in a very high quality. Thus, I would like to recommend its acceptance by Nature Communications in its current form.

Reply:

Thank the reviewer for the highly positive remarks and support of publication of this work.